# C-Reactive Protein-to-Albumin Ratio as a Predictive Indicator for Evaluating Tolerability in S-1 Adjuvant Chemotherapy after Curative Surgery for Pancreatic Cancer: An External Validation Cohort Study

**DOI:** 10.3390/cancers16193372

**Published:** 2024-10-01

**Authors:** Naotake Funamizu, Shozo Mori, Akimasa Sakamoto, Miku Iwata, Mikiya Shine, Chihiro Ito, Mio Uraoka, Yoshitomo Ueno, Kei Tamura, Yuzo Umeda, Taku Aoki, Yasutsugu Takada

**Affiliations:** 1Department of Hepato-Biliary Pancreatic and Transplantation Surgery, Ehime University Graduate School of Medicine, Shitsukawa 454, Toon 791-0295, Ehime, Japan; sakamoto.akimasa.kw@ehime-u.ac.jp (A.S.); miku.nkgw@gmail.com (M.I.); shine.mikiya.kz@ehime-u.ac.jp (M.S.); ito.chihiro.wk@ehime-u.ac.jp (C.I.); uraoka.mio.lr@ehime-u.ac.jp (M.U.); ueno.yoshitomo.xr@ehime-u.ac.jp (Y.U.); k-tamura@m.ehime-u.ac.jp (K.T.); umeda.yuzo.oe@ehime-u.ac.jp (Y.U.); takada-yasutsugu@yamatokoriyama.jcho.go.jp (Y.T.); 2Department of Hepato-Biliary Pancreatic Surgery, Dokkyo Medical University, Kitakobayashi 880, Mibu, Shimotsugagun 321-0293, Tochigi, Japan; shozomori@hotmail.co.jp (S.M.); aoki-2su@dokkyomed.ac.jp (T.A.)

**Keywords:** C-reactive protein-to-albumin ratio, S-1, adjuvant chemotherapy, pancreatic cancer, validation study

## Abstract

**Simple Summary:**

Adjuvant chemotherapy (AC) using S-1 has demonstrated favorable outcomes for patients with pancreatic cancer (PC). However, the current S-1 completion rate is insufficient to achieve its benefits. Moreover, an absence of dependable markers to forecast S-1 completion warrants C-reactive protein-to-albumin ratio (CAR) investigation as a potential predictive factor related to nutritional status. Previously, we revealed that a postoperative CAR value of ≥0.05 serves as a marker predicting S-1 AC treatment non-completion due to adverse events (AEs) in the Ehime study. Thus, this study aims to substantiate the correlation between postoperative CAR and S-1 therapy non-completion due to AEs using an alternative cohort from another institution (the Dokkyo study).

**Abstract:**

Background: S-1 in adjuvant chemotherapy (AC) administration after pancreatic cancer (PC) surgery has been standardized in Japan. The Ehime study confirmed that a postoperative higher C-reactive protein-to-albumin ratio (CAR) value predicted the risk of adverse event (AE)-related S-1 non-completion as an AC in patients with PC after curative surgery. This study aimed to investigate the index to predict S-1 tolerance among patients who underwent curative surgery for PC (the Dokkyo study). Methods: This retrospective validation cohort study included 172 patients at the Department of Hepato-Biliary Pancreatic Surgery, Dokkyo Medical University, Japan, from January 2010 to December 2022. All patients underwent nutritional screening using the postoperative CAR. S-1 completion status and its effect on prognosis were systematically followed up and investigated. We conducted a statistical analysis of predictive markers to investigate their association with S-1 completion. Results: Patients were categorized into the S-1 completion (N = 91) and non-completion (N = 81) groups. The S-1 completion group demonstrated a significantly lower CAR than the S1 non-completion group. Moreover, the current study revealed a significant difference in the S-1 completion rate, applying the CAR cutoff value of 0.05 established in the Ehime study. Additionally, univariate and multivariate analyses confirmed that a CAR of <0.05 was significantly associated with S-1 completion. Conclusions: The Dokkyo study confirmed the results observed in the Ehime study. Consequently, an increased postoperative CAR value appeared as a universal applicable marker for the risk factor of AE-related S-1 non-completion after curative surgery for patients with PC.

## 1. Introduction

Pancreatic cancer (PC), with a 5-year survival rate of approximately 10% in the United States, is increasingly known as a significant contributor to cancer-related fatalities. Additionally, approximately 80–85% of patients present with either unresectable or metastatic disease [1]. Surgical resection appears as the standard treatment, and adjuvant chemotherapy (AC) advancement has improved the long-term prognosis for individuals with this condition [2]. Several clinical trials have demonstrated convincing evidence indicating that, in resectable PC cases, the combined approach of radical resection and AC yields a markedly improved prognosis compared to surgery alone [3,4]. Particularly, the Japanese guidelines advocate for the standard use of oral S-1 administration as part of the AC protocol after radical surgery for patients with PC [5]. This recommendation is rooted in the outcomes of a randomized trial conducted by the Japan Adjuvant Study Group of Pancreatic Cancer (JASPAC01), indicating that patients treated with S-1 demonstrated significantly extended 5-year and median survival rates compared to those subjected to gemcitabine after radical surgery [6]. Accomplishing successful completion poses challenges due to the occurrence of postoperative complications (POCs) or adverse events (AEs) associated with the AC regimen itself despite the well-established efficacy of AC in ameliorating the prognosis of patients with PC [7].

Previous reports, including our institutional data, reported an S-1 therapy completion rate of approximately 70% [6,8]. Thus, promptly identifying and mitigating the risk factors associated with S-1 non-completion are clinically imperative to prevent unfavorable outcomes.

Recently, the C-reactive protein (CRP)-to-albumin ratio (CAR) has gained popularity for evaluating a patient’s nutritional status and predicting POCs and their outcomes [9,10,11,12]. Importantly, the CAR is easily accessible and cost-effective, requiring only CRP and serum albumin level data. A previous investigation (the Ehime study) revealed that a postoperative CAR of ≥0.05 could serve as a predictor for S-1 non-completion due to AEs in patients undergoing curative surgery for PC, considering the close correlation between postoperative nutritional status and S-1 therapy completion [13]. Therefore, the present Dokkyo study aims to confirm the association between postoperative CARs and S-1 tolerance in patients who underwent curative surgery for PC using an external cohort with the same CAR cutoff value (0.05). This is crucial as determining predictive markers for S-1 non-completion caused by AEs may help identify high-risk patients.

## 2. Materials and Methods

### 2.1. Patients

This retrospective external validation study included 301 consecutive patients who underwent radical pancreatic resection for PC at Dokkyo Medical University Hospital from January 2010 to December 2022. Significantly, this study excluded 129 patients because they did not commence S-1 AC due to variations in patient preferences (N = 69) and used different regimens (N = 60) (Figure 1).

The ultimate analysis involved a cohort of 172 patients. Notably, patient mortality was not observed within the initial 90 days postoperatively. The investigation entailed a comprehensive assessment of patients’ medical records, encompassing the retrieval of information associated with patients’ demographics, perioperative laboratory outcomes, clinical details during the perioperative period, pathological results, and postoperative prognoses.

Moreover, recognizing that the study protocol underwent thorough scrutiny and obtained approval from the institutional ethics committee of Ehime University Hospital (No. EUH2406008) is imperative, adhering to the principles of the Declaration of Helsinki as revised in 2013. Every participant, including retrospectively registered patients or their guardians, explicitly provided informed consent for the use of their medical data for scientific research endeavors.

### 2.2. Operative Procedures and Perioperative Management in Dokkyo Medical University Hospital

The predominant method for pancreatic anastomoses to the alimentary tract involved using the end-to-side pancreatojejunostomy technique during pancreatoduodenectomy. Three closed suction drainage tubes were inserted as a standard practice. The resection of the pancreas was primarily achieved using a linear stapler in instances of distal pancreatectomy. The surgeon’s preference identified the placement of two or three closed suction drainage tubes. The Clavien–Dindo (CD) classification was used to evaluate POCs, with complications of grade ≥3 considered major POCs [14].

### 2.3. AC Regimens and Postoperative Surveillance

Dokkyo Medical University Hospital scheduled AC to commence promptly after hospital discharge and continue for 6 months. The majority of patients started AC within 3 months of undergoing curative surgery. The treatment protocol involved oral S-1 administration, with dosages of 80–120 mg based on body surface area, taken twice a day for 28 days, followed by a 14-day rest period. This treatment cycle was repeated for 6 months unless intolerable toxicity appeared.

The relative dose intensity (RDI) was calculated as the ratio of the actual dose intensity to the standard or planned S-1 dose [15]. S-1 therapy completion was the consistent oral S-1 administration with an RDI of >80% [16]. Hematological and biochemical analyses, along with clinical parameters, such as body weight fluctuations, were assessed at AC initiation and during all subsequent appointments.

Postoperative surveillance included monthly blood tests and contrast-enhanced computed tomography scans performed every three months at Dokkyo Medical University Hospital. AEs were evaluated per the Common Terminology Criteria for Adverse Events version 5.0, with AEs of grade ≥3 categorized as severe AEs [17].

### 2.4. Definition of CAR

The CAR value was calculated from blood tests conducted postoperatively before the initiation of AC. The CAR was measured using the following formula: CAR = [CRP (mg/dL)]/[albumin (g/dL)] [13,18]. Subsequently, patients were categorized based on the determined CAR threshold value (0.05) into the S-1 completion and non-completion groups.

### 2.5. Statistical Analysis

The Statistical Package for the Social Sciences (SPSS^®^) version 16.0 for Windows^®^ (SPSS, Chicago, IL, USA) and GraphPad Prism version 5.0 (GraphPad Software Inc., La Jolla, CA, USA) were used for all statistical analyses. Patient demographics are presented as the medians and interquartile ranges for nonparametric distributions, whereas categorical data are expressed as numbers and percentages. The χ^2^ test, Fisher’s exact test, and the U test were used to determine the statistical significance for patient demographics and outcomes, as appropriate. Univariate and multivariate analyses were conducted to identify independent factors affecting S-1 completion. The Ehime study-derived cutoff value was used to identify the optimal cutoff value for CAR in predicting the risk of S-1 non-completion, although receiver operating characteristic (ROC) curve analysis was conducted. Additionally, the cutoff values for each variable in multivariate analysis were selected by ROC analysis. Overall survival (OS) and recurrence-free survival (RFS) after curative surgery were evaluated using the Kaplan–Meier method, and survival curves were compared using the log-rank test. A probability level of *p*-values of <0.05 was considered statistically significant.

## 3. Results

### 3.1. Patient Characteristics with or without S-1 Completion in the Dokkyo Cohort

Curative surgery was performed on 172 patients with PC, followed by S-1 initiation as the AC regimen in the study period. Of these patients, 91 (52.9%) successfully continued with AC, maintaining an RDI of >80%. Conversely, 38 patients experienced dose reduction or treatment interruption caused by AEs (Figure 1).

Table 1 shows a comprehensive overview of the detailed patient characteristics. Furthermore, Table 2 outlines the laboratory test results at the onset of AC, along with relevant AC-related factors. The AC duration and tumor marker levels, such as carbohydrate antigen 19-9 (CA19-9) and carcinoembryonic antigen (CEA), demonstrated no variations between the S-1 completion and non-completion groups. However, noteworthy distinctions were determined in the albumin, CRP, CAR, and severe AEs between the groups, with statistical significance observed for those variables (*p* < 0.001).

### 3.2. Calculation of an Optimal CAR

The optimal cutoff value was determined using data from the Ehime study. Moreover, the ROC curve analysis indicated that the areas under the curves for the CAR, albumin, and CRP levels were 0.77, 0.68, and 0.69, respectively. Additionally, when the CAR cutoff value is set at 0.05, this value exhibited a sensitivity of 71.4%, specificity of 76.3%, and likelihood ratio of 3.01 (Figure 2). When patients were categorized by S-1 completion status, they were divided into two groups based on the postoperative CAR cutoff value: the higher CAR group (CAR of ≥0.05, *n* = 49) and the lower CAR group (CAR of <0.05, *n* = 80). S-1 non-completion was observed in 23 patients (46.9%) in the higher CAR group and 15 patients (18.8%) in the lower CAR group. Moreover, in all patients who underwent S-1 AC therapy, patients were categorized into two groups based on the CAR cutoff value: the higher CAR group (CAR of ≥0.05, *n* = 80) and the lower CAR group (CAR of <0.05, *n* = 92). S-1 non-completion occurred in 54 (67.5%) and 27 (29.3%) patients in the higher and lower CAR groups, respectively. Univariate analysis was conducted to evaluate a postoperative CAR value of ≥0.05 as a risk factor for S-1 treatment non-completion in both groups, respectively (*p* < 0.001).

The area under the curve of the C-reactive protein (CRP)-to-albumin ratio, CRP, and albumin value were compared by receiver operating characteristic curve analysis. With a CAR cutoff value of ≥0.05, the sensitivity and specificity were determined to be 71.4% and 76.3%, respectively.

### 3.3. Multivariate Analysis for S-1 Completion

All predictive factors that correlated with S-1 completion in the univariate analysis were entered into the multivariate analysis (Table 3). Multivariate analysis identified a CAR of <0.05 as an independent risk factor for S-1 completion (hazard ratio [HR]: 5.14; 95% confidence interval [CI]: 2.01–13.30; *p* = 0.001).

### 3.4. CAR and Patient Outcome

The prognostic value of a CAR of <0.05 was investigated (Figure 3). Patients with a CAR of ≥0.05 had worse RFS (95% CI: 0.418–0.859, HR: 0.599, *p* = 0.005) and OS (95% CI: 0.342–0.747, HR: 0.506, *p* < 0.001) than those with a CAR of <0.05. Additionally, in the entire cohort of patients who received S-1 as AC (*n* = 172), a CAR > 0.05 was also shown to be a significant prognostic factor (Figure 4).

## 4. Discussion

Recent research results have revealed that nutritional status significantly affects perioperative management. Previous reports have revealed that nutritional indices, such as the geriatric nutritional risk index (GNRI), lymphocyte-to-neutrophil ratio, prognostic nutritional index (PNI), and the CAR, are frequently related to POCs, such as postoperative pancreatic fistula and surgical site infection, and prognosis in various cancer types [19,20,21,22,23,24].

Regarding the association between chemotherapy and nutritional status, randomized controlled trials revealed that nutritional status improvement is statistically correlated with the response rate or AEs during neoadjuvant AC (NAC) in patients with breast cancer [25,26]. Additionally, nutritional status has been associated with the toxicity and efficacy of chemotherapy, including NAC, not only in breast cancer but also in gastrointestinal and head and neck cancers [27,28]. Furthermore, the risk of discontinuing chemotherapy may be influenced by the nutritional status observed before chemotherapy initiation [29]. Moreover, profound malnutrition and substantial weight loss demonstrated a significant association with treatment effectiveness [30]. Recent years have witnessed a heightened concentration on research investigating the correlation between nutritional status and chemotherapy, with a notable surge in reported studies. However, attitudes toward nutritional care among surgeons, oncologists, and nutritionists were divergent despite the several reported benefits of nutritional status for cancer patients. A considerable proportion of patients do not undergo nutritional assessment or receive nutritional treatment [31]. Albaro et al. revealed that more than one-third of patients with cancer initiating chemotherapy are candidates for early nutritional intervention [32]. Hence, a demand has arisen for presenting higher-level evidence exhibiting the use of assessing the nutritional status of patients with cancer.

Among chemotherapy, even when limited to AC therapy, more recent studies have revealed that nutritional status affects the efficacy and completion rates of AC [33,34,35]. Numerous nutritional parameters have been investigated concerning AC tolerance with S-1 across various cancer types. These include parameters such as the GNRI, PNI, and neutrophil-to-lymphocyte ratio [8,12,36,37]. Therefore, amidst the reported associations between diverse nutritional statuses and various indicators, we speculated about the potential effects of the CAR on S1 AC completion rates for patients with PC. Our Ehime study confirmed the association between the CAR and S-1 completion rate for the first time [13]. We conducted a validation study using an external cohort (the Dokkyo study) to fortify the significance of the previous results.

S-1, an oral anticancer drug, is formulated with tegafur, serving as a prodrug for three chemotherapy components, including fluorouracil, gimeracil, and oteracil potassium. Gimeracil functions by inhibiting dihydropyrimidine dehydrogenase activity, causing heightened fluorouracil levels in both the bloodstream and tumor tissue. Conversely, oteracil potassium assumes a vital role in suppressing fluorouracil phosphorylation within the gastrointestinal tract, thereby effectively lowering the potential for gastrointestinal toxicity [38]. S-1 has demonstrated robust efficacy in conferring survival benefits when administered as AC for Japanese patients diagnosed with several cancer types, including in the PC JASPAC01 trial [6] and biliary tract cancer [39]. Thus, S-1 has been established as beneficial for postoperative AC in Japanese patients with PC.

However, clinical studies indicated a current challenge where some postoperative patients with PC may find it difficult to continue oral S-1 intake due to AEs or POCs. The preceding test Ehime cohort indicated an S-1 completion rate of 69.5% [13]. Likewise, the Dokkyo validation study revealed that 70.5%% of patients completed the AC treatment, consistent with earlier observations. The identification of a non-invasive and straightforward marker for predicting S-1 non-completion due to AEs is deemed highly crucial, considering the failure rate of treatment nearing 30%. This would serve as a valuable means to evaluate patient prognosis and identify the intervals for postoperative follow-up.

The results of the Dokkyo study confirmed the correlation between a CAR value of >0.05 and an increased likelihood of S-1 therapy non-completion due to AEs. This indicates that improving the nutritional or inflammatory status before AC may mitigate the risk of S-1 treatment non-completion. Additionally, both cohorts have indicated the CAR as a predictive factor for S-1 therapy completion, and consequently, it may also function as a prognostic factor. Noteworthily, a CAR of ≥0.05 corresponds to the CAR value determined as a predictive factor for POCs in a previous study that involved patients with PC from our institution [9,10]. This consistency emphasizes the potential significance of CAR in both POCs and S-1 therapy non-completion due to AEs [9,10,13]. Nutritional indices, including CAR, have been deeply associated not only with PC but also with prognostic factors, POC prediction, and chemotherapy AEs in various cancers, as evidenced by reviews and meta-analyses [40,41,42,43,44,45]. However, scattered reports indicated no association between nutritional status and prognosis, emphasizing the need for validation through forthcoming large-scale studies [46,47].

However, this study has notable limitations. First, this is a single-center study with a relatively modest dataset to validate our previous results, which could have introduced bias into the data analysis, thereby restricting a comprehensive assessment of the effect of the CAR. Second, the retrospective study design may have introduced selection bias. Third, S-1 treatment aspects, such as the initiation timing, dosage, dose reduction, and withdrawal, were determined at the attending physician’s discretion. Consequently, validating the results of this study through a more extensive and diverse series is imperative.

## 5. Conclusions

In the present study, surpassing a postoperative CAR of ≥0.05 not only markedly diminishes the S-1 as AC non-completion rate due to AEs compared to values of <0.05 but also manifests itself as a substantial adverse prognostic factor.

## Figures and Tables

**Figure 1 cancers-16-03372-f001:**
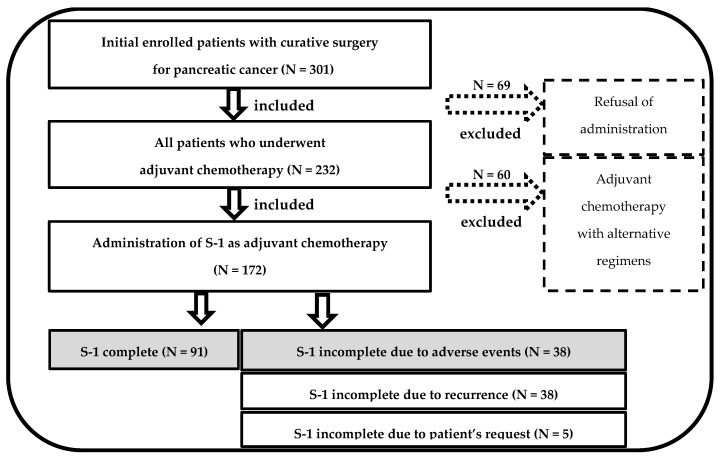
Flowchart for patient selection.

**Figure 2 cancers-16-03372-f002:**
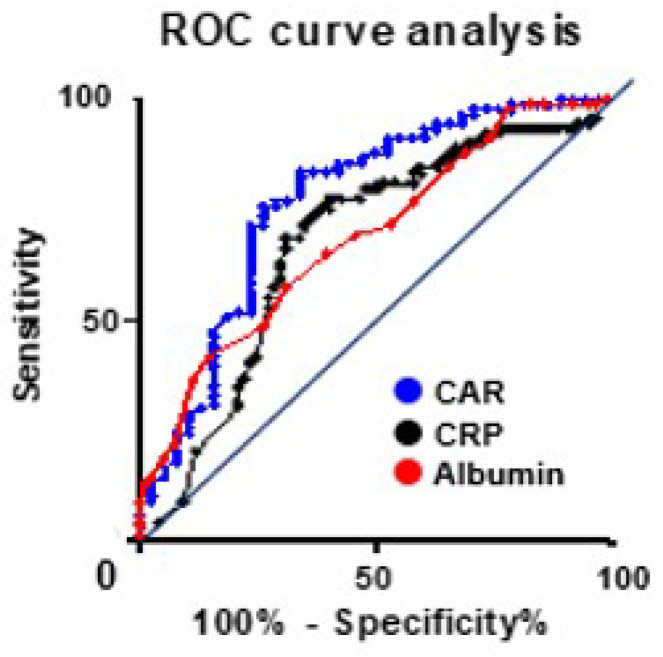
ROC curve analysis.

**Figure 3 cancers-16-03372-f003:**
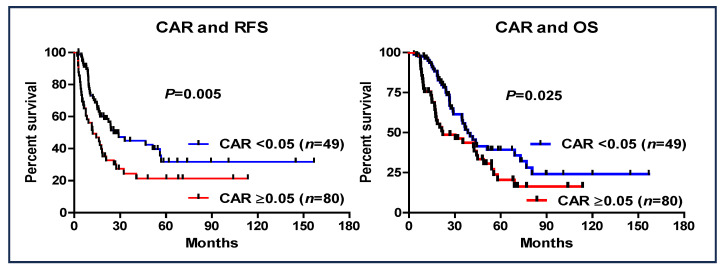
Recurrence-free survival and overall survival based on the cutoff value of CAR of ≥0.05 in patients with pancreatic cancer (S-1 complete group vs. S-1 non-complete group).

**Figure 4 cancers-16-03372-f004:**
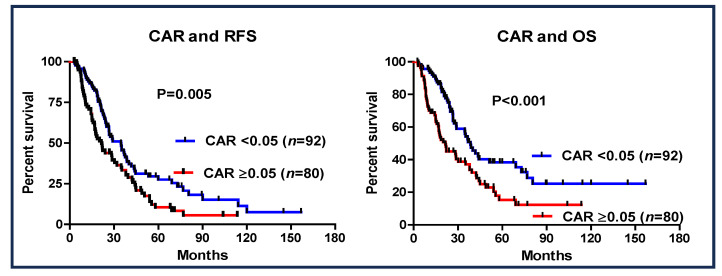
Recurrence-free survival and overall survival based on the cutoff value of CAR of ≥0.05 in patients with pancreatic cancer who underwent S-1 AC therapy.

**Table 1 cancers-16-03372-t001:** Comparison of perioperative factors between the S1 completion and non-completion groups.

Patient Characteristics	S-1 Complete Group(*n* = 91)	S-1 Non-Complete Group (*n* = 38)	*p*-Value
Sex (male rate%)	52 (57.1%)	17 (44.7%)	0.198
Age (year)	65.8 (43–81)	68.7 (47–84)	0.127
Body mass index (kg/m^2^)	22.9 ± 0.4	22.6 ± 0.7	0.736
Neoadjuvant chemotherapy, n (%)	48 (52.7%)	21 (55.3%)	0.794
Operation methods			
DP	32 (35.2%)	9 (23.7%)	
PD	54 (59.3%)	27 (71.1%)	
TP	5 (5.5%)	2 (5.3%)	
Operation time (min)	484.8 ± 15.1	525.9 ± 22.8	0.14
Estimated blood loss (mL)	746.7 ± 50.7	883.6 ± 109.6	0.197
CD classification over grade 3	23 (25.3%)	13 (34.2%)	0.302
Postoperative hospital stays (days)	39.1 ± 2.0	31.7 ± 3.4	0.499
Pathological Stage			
1	12 (13.2%)	2 (5.3%)	
2	77 (84.6%)	29 (76.3%)	
3	2 (2.2%)	7 (18.4%)	

DP: distal pancreatectomy; PD: pancreatoduodenectomy; TP: total pancreatectomy.

**Table 2 cancers-16-03372-t002:** Comparison of factors before and after AC between the S1 completion and non-completion groups.

Variables	S-1 Complete Group (*n* = 91)	S-1 Non-Complete Group (*n* = 38)	*p*-Value
Duration to AC initiation (day)	63.1 ± 3.8	67.4 ± 8.7	0.422
Data at the initiation of AC			
Body mass index (kg/m^2^)	20.9 ± 0.4	20.4 ± 0.7	0.536
Alb (mg/dL)	3.5 ± 0.1	3.0 ± 0.1	<0.001
CRP (mg/dL)	0.2 ± 0.1	1.0 ± 0.2	<0.001
CEA (ng/mL)	2.9 ± 0.3	4.4 ± 1.0	0.062
CA19-9 (U/mL)	69.2 ± 18.4	403.3 ± 431.6	0.123
CAR	0.07 ± 0.02	0.25 ± 0.04	<0.001
Severe AEs	6 (6.6%)	19 (50.0%)	<0.001

Alb: albumin; CRP: c-reactive protein; CEA: carcinoembryonic antigen; CA19-9: carbohydrate antigen 19-9; CAR: CRP-to-albumin ratio; AEs: adverse event.

**Table 3 cancers-16-03372-t003:** Multivariate analysis of factors associated with S1 non-completion.

Multivariate Analysis for S-1 Non-Completion		
	Hazard Ratio (95% CI)	*p*-Value
CEA > 2.4	1.64 (0.65–4.11)	0.295
Severe AEs	7.20 (2.45–21.10)	<0.001
CAR ≥ 0.05	5.18 (2.01–13.30)	<0.001

AEs: adverse events; CAR: CRP/albumin ratio.

## Data Availability

The datasets used and analyzed during the current study are available from the corresponding author upon reasonable request.

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
