# Peer review of "C-Reactive Protein-to-Albumin Ratio as a Predictive Indicator for Evaluating Tolerability in S-1 Adjuvant Chemotherapy after Curative Surgery for Pancreatic Cancer: An External Validation Cohort Study"

_cancers, 2024, doi:10.3390/cancers16193372_

Round 1

Reviewer 1 Report

Comments and Suggestions for Authors

The manuscript "C-reactive protein-to-albumin ratio as a predictive indicator for evaluating tolerability in S-1 adjuvant chemotherapy after curative surgery for pancreatic cancer" validates the use of the C-reactive protein-to-albumin ratio (CAR) as a marker for predicting S-1 chemotherapy non-completion due to adverse events in pancreatic cancer patients. 

The study builds on previous findings from the Ehime study and confirms the association between postoperative CAR and chemotherapy tolerance using a cohort from Dokkyo Medical University. The study analyzed 172 patients retrospectively, calculating CAR postoperatively to group patients based on S-1 completion or non-completion. CAR was significantly associated with non-completion, and higher CAR was linked to worse recurrence-free and overall survival. Multivariate analysis confirmed CAR as an independent risk factor for non-completion.

The manuscript effectively replicates previous findings, showing consistency in using CAR as a predictive marker. The methodology is clear, and statistical analyses support the conclusions. 

The study addresses a relevant clinical issue, offering potential improvements in adjuvant chemotherapy outcomes for pancreatic cancer.

However, external validity is limited due to the single-center data. Broader validation with larger, multi-center datasets is recommended to solidify CAR as a universally applicable marker in the future.

Minor point:

Figure 4: The CAR values are currently displayed as either greater or smaller than 0.05. The inclusion of CAR = 0.05 needs to be integrated.

Recommendation: Accept with minor revisions.

Author Response

Reviewer1:

The manuscript "C-reactive protein-to-albumin ratio as a predictive indicator for evaluating tolerability in S-1 adjuvant chemotherapy after curative surgery for pancreatic cancer" validates the use of the C-reactive protein-to-albumin ratio (CAR) as a marker for predicting S-1 chemotherapy non-completion due to adverse events in pancreatic cancer patients. 

The study builds on previous findings from the Ehime study and confirms the association between postoperative CAR and chemotherapy tolerance using a cohort from Dokkyo Medical University. The study analyzed 172 patients retrospectively, calculating CAR postoperatively to group patients based on S-1 completion or non-completion. CAR was significantly associated with non-completion, and higher CAR was linked to worse recurrence-free and overall survival. Multivariate analysis confirmed CAR as an independent risk factor for non-completion.

The manuscript effectively replicates previous findings, showing consistency in using CAR as a predictive marker. The methodology is clear, and statistical analyses support the conclusions. 

The study addresses a relevant clinical issue, offering potential improvements in adjuvant chemotherapy outcomes for pancreatic cancer.

However, external validity is limited due to the single-center data. Broader validation with larger, multi-center datasets is recommended to solidify CAR as a universally applicable marker in the future.

Minor point:

Figure 4: The CAR values are currently displayed as either greater or smaller than 0.05. The inclusion of CAR = 0.05 needs to be integrated.

#Thank you for your input. I will make the correction.

Recommendation: Accept with minor revisions.

Reviewer 2 Report

Comments and Suggestions for Authors

This is an interesting paper, which intends to validate a previous study (Ehime) where C reactive protein to albumin radio may be used as a predictive index to evaluation tolerability in an adjuvant chemotherapy setting (S-1-based) after curative surgery for patients with pancreas cancer.

While the present study (Dokkyo) is well justified and performed with results supporting the conclusions of the initial study, there are many questions unanswered or not discussed to convince the reader about the utility/reliability of this predictive index.

Is indeed the prediction of S-1 treatment non-completion due to the upcoming adverse events or is actually predicting a potential different degrees of resistance against the drug? This may translate to different CRP and albumin levels since it suggests a more aggressive cancer secreting different molecules (like cachectin) which may trigger sickness, including adverse events-like symptoms?

Co-morbidities and their corresponding treatments may significantly change CRP and albumin levels. Why the authors did not include these critical features? I believe that it is mandatory for this kind of study to perform a statistical adjustment to eliminating critical confounding factors. Nevertheless, co-morbidities and their treatments are also well known as “sensitizers” for the cancer chemotherapy-induced adverse effects.

To validate this predictive index, the authors should have applied the same strategy for other cancers where S-1 is given as an adjuvant treatment after curative surgery. If this index applies only for pancreatic cancer, how the authors see the interaction between drug, cancer cells, and general body physiology (on one hand) and the induction of adverse events (on the other hand)?

It will be interesting to see what type of adverse events were induced followed a separate analysis to determine if the CRP/albumin index may be a better predictor for some certain adverse event(s)

It appears that only the baseline CRP/albumin index (before adjuvant chemotherapy) was used for the stats, despite being measured throughout all AC treatment. Did the authors compare the dynamics of the index between the two groups at multiple time points?

Since the study was performed on a pool of patients investigated within 12 years, were the blood tests performed under the same methodologies, with the same range of normal values?

Finally, can the authors discuss if this may be used for other types of adjuvant chemotherapies for pancreatic (or other) cancers.

As a minor note, please fix the typo in the Figure 1. N=231 must be written correctly as N=232.

Author Response

Reviewer2:

This is an interesting paper, which intends to validate a previous study (Ehime) where C reactive protein to albumin radio may be used as a predictive index to evaluation tolerability in an adjuvant chemotherapy setting (S-1-based) after curative surgery for patients with pancreas cancer.

While the present study (Dokkyo) is well justified and performed with results supporting the conclusions of the initial study, there are many questions unanswered or not discussed to convince the reader about the utility/reliability of this predictive index.

  1. Is indeed the prediction of S-1 treatment non-completion due to the upcoming adverse events or is actually predicting a potential different degrees of resistance against the drug? This may translate to different CRP and albumin levels since it suggests a more aggressive cancer secreting different molecules (like cachectin) which may trigger sickness, including adverse events-like symptoms?

#Thank you for the excellent question. As the reviewer pointed out, the inability to complete S-1 treatment could indeed be due to tumor resistance or cytokines associated with more advanced and aggressive cancers. However, since S-1 is primarily a drug for recurrence prevention, we have excluded cases of treatment discontinuation due to disease progression from this study in order to minimize the influence of disease progression as a factor.

  1. Co-morbidities and their corresponding treatments may significantly change CRP and albumin levels. Why the authors did not include these critical features? I believe that it is mandatory for this kind of study to perform a statistical adjustment to eliminating critical confounding factors. Nevertheless, co-morbidities and their treatments are also well known as “sensitizers” for the cancer chemotherapy-induced adverse effects.

# Thank you for your feedback. As the reviewer pointed out, co-morbidities and their treatments can significantly affect CRP and albumin levels. Fortunately, patients with inflammatory conditions such as autoimmune diseases and COPD were not included in this study. Additionally, CAR was measured postoperatively, and in cases of postoperative inflammation such as cholangitis or abscess, S-1 treatment was delayed until the inflammation resolved. Therefore, we believe that the impact of confounding factors in this study is minimal.

  1. To validate this predictive index, the authors should have applied the same strategy for other cancers where S-1 is given as an adjuvant treatment after curative surgery. If this index applies only for pancreatic cancer, how the authors see the interaction between drug, cancer cells, and general body physiology (on one hand) and the induction of adverse events (on the other hand)?

#Thank you for your valuable feedback. As we specialize in hepatobiliary and pancreatic surgery, we do not have the means to investigate other cancer types, such as gastric cancer, where S-1 is used as adjuvant therapy. However, S-1 is expected to be used as adjuvant therapy for biliary tract cancer in the future, and we plan to validate our findings in that context as well. The second issue you raised is very important, but it is a difficult question. As mentioned in our discussion, there are reports suggesting that weight loss is associated with the difficulty of continuing S-1 treatment, and we believe that worsening nutritional status may enhance the adverse effects.

  1. It will be interesting to see what type of adverse events were induced followed a separate analysis to determine if the CRP/albumin index may be a better predictor for some certain adverse event(s)

#Thank you for the valuable suggestion. We will aim to gather more cases and report which adverse events are most strongly associated with the CRP/albumin index.

  1. It appears that only the baseline CRP/albumin index (before adjuvant chemotherapy) was used for the stats, despite being measured throughout all AC treatment. Did the authors compare the dynamics of the index between the two groups at multiple time points?

#As mentioned earlier, there were cases with elevated inflammatory responses due to postoperative complications, so AC therapy was not initiated until these improved. Therefore, the CAR can vary significantly based on the presence or absence of complications, and calculations are based on blood test results obtained before the initiation of AC.

  1. Since the study was performed on a pool of patients investigated within 12 years, were the blood tests performed under the same methodologies, with the same range of normal values?

#The normal range has not changed.

  1. Finally, can the authors discuss if this may be used for other types of adjuvant chemotherapies for pancreatic (or other) cancers.

#In Japan, the types of cancers for which AC therapy can be used with insurance coverage are limited. In pancreatic cancer, other types include acinar cell carcinoma and neuroendocrine carcinoma (NEC), but these are not indicated for S-1 adjuvant therapy. For gastric cancer, S-1 is administered for one year as postoperative adjuvant therapy; however, since we specialize in hepatobiliary and pancreatic surgery, we do not have data on gastric cancer. Therefore, our response is that we cannot engage in a discussion on this topic.

  1. As a minor note, please fix the typo in the Figure 1. N=231 must be written correctly as N=232.

#Thank you for your feedback. I will make the correction.

Round 2

Reviewer 2 Report

Comments and Suggestions for Authors

I would like to thank the authors for their concise and pertinent responses. In my opinion, the manuscript is suitable for publication in the present form.